# Alkaline Treatment Investigation for Sedge Fibers (*Cyperus malaccensis*): A Promising Enhancement

**DOI:** 10.3390/polym15092153

**Published:** 2023-04-30

**Authors:** Lucas de Mendonça Neuba, Raí Felipe Pereira Junio, Andressa Teixeira Souza, Yago Soares Chaves, Sheron Tavares, Alexandre Alvarenga Palmeira, Sergio Neves Monteiro, Artur Camposo Pereira

**Affiliations:** 1Department of Materials Science, Military Institute of Engineering, Rio de Janeiro 22290-270, RJ, Brazil; lucasmneuba@ime.eb.br (L.d.M.N.); raivsjfelipe@hotmail.com (R.F.P.J.); andressa.souza@ime.eb.br (A.T.S.); yago_soares2@ime.eb.br (Y.S.C.); sergio.neves@ime.eb.br (S.N.M.); 2Materials Science and Engineering Program, University of California San Diego, 9500 Gilman Drive, La Jolla, CA 92093, USA; sstavare@eng.ucsd.edu; 3Department of Mechanical Engineering, University of Rio de Janeiro State, Resende 27537-000, RJ, Brazil; alex.a.palmeira@gmail.com

**Keywords:** alkaline treatment, sedge fibers, physical characterization, polymer reinforcement

## Abstract

Natural fibers have some advantages in comparison to synthetic fibers, especially because they are more environmentally friendly. For this reason, using them as a reinforcement for polymeric matrices is growing exponentially. However, they present the disadvantage of having the hydrophilic nature, which strongly reduces the interface interaction. Sedge fibers have been investigated when reinforcing an epoxy matrix in terms of ballistic properties and mechanical performance. Aiming to enhance the fiber−matrix interface, an alkali treatment was proposed. The group conditions were divided into three NaOH concentrations (3%, 5%, and 10%), as well as the three periods of immersion (24, 48, and 72 h). Therefore, nine different conditions were investigated in terms of their thermal behaviors, chemical structures, physical structures, and morphological aspects. Based on TGA curves, it could be noticed that treatments related to 3% NaOH for 24 h and 48 h exhibited better thermal stability properties. For the time of 48 h, better thermal stability with for a decay of the thermal DSC curve was shown for all treatment conditions. The FTIR spectra has shown a reduction of waxes for higher NaOH concentrations. The XRD diffractogram exhibited an increase in the crystallinity index only for 5% NaOH and an immersion time of 48 h. The morphological aspects of fibers treated with 5% and 10% of NaOH have shown that the treatments have damaged the fiber, which highlighted the crystallinity index reductions.

## 1. Introduction

The processing of polymeric matrices reinforced by synthetic fibers requires a great amount of energy. It also displays a harmful effect on the environment generated by pollution. The concern associated with the disadvantages and negative impacts of using these materials has increased. As a consequence, researchers focused their efforts on the past decades to apply natural lignocellulosic fibers (NLFs) as a reinforcement. The reason was due to the distinct characteristics they present. Among the advantages presented by these biological materials over synthetic fibers, it is worth mentioning the easiness of processing, non-toxicity, less abrasiveness, and full biodegradation can reduce the carbon foot of the composite materials and possess relatively high specific mechanical properties [1,2,3,4]. Additionally, the composites acquire an elevated moisture absorption characteristic, which reduces their use in several applications [5,6].

Therefore, the employment of NLFs as a reinforcement for thermoplastic or thermoset polymeric resins has become relevant in building [7,8], automotive [9,10], and military [11,12,13] sectors. For this biomaterial, a single natural fiber is considered a natural composite. Due to lignin and hemicellulose acting as a glue that links microfibrils, the latter microstructure is the reinforcement phase [14,15]. However, these biological materials present a concerning drawback related to the fiber−matrix interface being considered weak. It is mainly occasioned by the difference between the hydrophobic nature of the polymer matrix set and the hydrophilic nature of the natural fiber. The hydrophilic aspect of NLFs is particularly induced by their microstructure that owns strongly polarized hydroxyl groups [16,17].

A strong interaction on the interface of composites is critical for them to disclose higher mechanical properties [18,19]. A possible resolution to enhancing the interaction between NLFs and polymeric matrices is physical or chemical treatments. Among the applied methods, one of the most commonly used is the alkaline or mercerization treatment. The meaningful modification conducted by this alkaline treatment is the hydrogen bond rupture in the structure, bringing an increase in the superficial roughness [20,21]. Thus, it plays a positive role in the mechanical interlocking between the fiber and the matrix. Moreover, it can remove certain amounts of wax, oils, and lignin that cover the cell, as well as exposing short-length crystallites and depolymerizes cellulose [22,23]. Thus, it directly influences the degree of polymerization, cellulosic fibrils, and partial removal of lignin and hemicellulose. It also increases the percentage of cellulose exposed on the surface, resulting in the growth of reactive sites [24,25].

This treatment consists of immersing fibers in an aqueous NaOH solution, previously establishing the concentration and the immersion time. Nevertheless, a high alkaline concentration or period of exposure excessively increases NLFs delignification, which leads to weaker or damaged fibers, as well as diminishing their mechanical performance [26]. For this reason, achieving an optimum value between the two parameters is essential to enhance fibers’ properties and consequently composites.

Several studies have related that chemical treatments, including alkaline, induce a decrease in the tensile strength of fibers. This might be assigned to damage present on the cell wall and the exceeding removal of lignin and hemicellulose that act as a matrix phase in NLFs structures [27,28,29]. Nonetheless, the thermal stability of fibers is significantly improved, as in the case of *Luffa cylindrica* which was immersed in a 2% NaOH alkaline solution or even the sisal [30]. It also reduces the moisture absorption for composites reinforced by NLFs; for example, jute fibers could considerably reduce water absorption, when they were treated with 5%, 10%, and 15% of NaOH [31]. Thus, it enables the use of these composites in several applications due to the fact that fibers develop a higher hydrophobicity. As a consequence, the interface of the composites is not affected, improving the performance of the composites [32,33].

The seven-islands-sedge (*Cyperus malaccensis*) or sedge for short discloses a relatively good Izod impact energy when applied as a reinforcement for an epoxy matrix [34]. However, the result regarding the tensile strength was not satisfactory. A possible reason for this behavior could be assigned to weak interfacial interaction, mainly caused by waxes and moisture absorption present on the surface of the sedge fibers. Therefore, the present study focused on investigating nine alkaline treatment conditions following NaOH concentrations of 3%, 5%, and 10% as well as immersion times of 24, 48, and 72 h. This present investigation aimed to be a remarkable enhancement for the ballistic, thermal, and mechanical performance already reported [35,36,37]. Thus, it could be a promising sustainable material for engineering and technological applications and present a significant reduction in the manufacturing cost for novel composites. It also should be very important for countries that cultivate this plant species, such as Brazil, since the use of sedge as a natural fiber can contribute to the circular economy concept. 

Thermogravimetric analysis (TGA) was performed to verify if a reduction of the moisture absorption and a thermal stability enhancement occurred. Furthermore, morphological analysis was conducted by scanning electron microscopy (SEM) to confirm if the surface roughness increased partial removal of the structural components. The Fourier-transform infrared spectroscopy (FTIR) analysis allowed checking the absence of the components in the chemical structure and possible interaction with Na. At last, an investigation of the fiber structure has been performed by the X-ray diffraction (XRD), where the crystallinity index, novel peaks related to cellulose, and the crystallite size were evaluated.

## 2. Materials and Methods

### 2.1. Materials and Processing

CM sedge fibers were extracted from as-received stalks acquired by a company known as Artevale in São Paulo, Brazil. The stalks were manually cleaned and then immersed in deionized water for 24 h, to provide greater flexibility and easier defibration. The manual extraction of fibers was performed using a sharp stiletto to carefully separate them from the stalks, which were originally obtained by the commercial supplier from the plant stem. The fiber bundle was cut to approximately 900 mm in length and dried in an oven at 70 °C for 24 h. These fibers were not subjected to any chemical treatment. Before the coating processing, the fiber bundle was cut to 150 mm.

Furthermore, the sedge fibers were subjected to NaOH alkaline treatment in different conditions, as shown in Table 1. The choice of the immersion time and NaOH concentration was based on the effectiveness of the treatment. Consequently, it may raise a condition in which the treatment caused serious degradation to the fiber integrity. Alkali solutions of 3%, 5%, and 10% (*w*/*v*) NaOH were prepared with distilled water. At first, the clean and dried sedge fibers were soaked and immersed in an alkali solution at room temperature for the proposed times without stirring. After 24, 48, or 72 h, the sedge fibers were then washed several times with distilled water to neutralize the remaining NaOH on the surface of the fibers and then dried at 80 °C in an oven for 24 h.

### 2.2. Thermogravimetric Analysis (TGA/DTG)

TGA was conducted to supplement previous results without NaOH treatment [38]. The sedge fibers were comminuted and inserted into a platinum crucible. The test was performed using a model DTG-60H (Shimadzu, Tokyo, Japan). The sample was heated from 25 to 600 °C at a rate of 10 °C/min under a nitrogen atmosphere and a gas flow of 50 mL/min. 

### 2.3. Differential Scanning Calorimetry (DSC)

DSC analyses of the composites were also performed to supplement previous results without NaOH treatment. The fibers were comminuted and placed in a platinum crucible. A model DTG-60H (Shimadzu, Tokyo, Japan) was used, with a heating rate of 10 °C/min starting at 30 up to 600 °C, on a nitrogen atmosphere with a gas flow of 50 mL/min.

### 2.4. Fourier-Transform Infrared Spectroscopy (FTIR)

Fourier-transform infrared spectroscopy (FTIR) analysis was performed on a model IRPrestige-21 (Shimadzu, Tokyo, Japan). The sedge fibers were grounded in the required powder condition to produce the KBr sample tablets. Then, they were scanned from 4000 to 400 cm^−1^. 

### 2.5. X-ray Diffraction (XRD) 

The crystallinity index (*I*_c_) and the crystallite size (CS) of the sedge fibers were calculated from the diffractogram obtained by X-ray diffraction (XRD). To calculate the fiber *I*_c_, the proposed method by Segal et al. [39] was used, in which the crystallinity is given by the difference of the peak relative to the (0 0 2) plane and the peak of the amorphous halo in the diffractogram:(1)$$I_{c}=\frac{I_{\;002}-I_{101}}{I_{002}}\times 100$$

A PANalytical X’pert Pro MRD diffractometer, with a cobalt anode (0.1789 nm), was used with the following parameters to obtain the diffractogram: a scan rate of 0.05 (2*θ*/s), scanning from 5° to 75°, a current of 40 mA, and a voltage of 40 kV. For the calculation of the CS, the method used was established by Scherer’s relations as expressed by the equation below [40]: (2)$$\mathrm{CS}=\frac{K\lambda }{\beta \cos\theta }\;$$
where *K* is the constant (0.89); *λ* corresponds to the intensity of radiation; *β* denotes the full-width at half-maximum (FWHM); and *θ* is the Bragg’s angle.

### 2.6. Scanning Electron Microscopy (SEM)

The sedge fiber surface was analyzed using a model FEI Apreo FE-SEM equipment (Waltham, MA, USA). The equipment operated with a secondary electron detector at an acceleration voltage between 9 and 10 kV. The fiber was platinum-coated in the equipment in order to create an electron current, so that images illustrating the mechanisms of failure could be observed with a higher resolution and few artifacts.

## 3. Results and Discussion

### 3.1. Thermogravimetric Analysis

Figure 1, Figure 2 and Figure 3 and Table 2 depict the results of TGA and first-order derivative analysis (DTG) for the sedge fibers treated with NaOH concentrations of 3%, 5%, and 10% for 24, 48, and 72 h.

It could be observed that the fiber mass losses for all treatments at temperatures up to 200 °C were around ~12%, due to the evaporation of the moisture content on the surface of the fibers. The untreated sedge fibers have shown a mass loss of 13.23% [38], which was higher than the mass losses of the fibers treated with 5% and 10% of NaOH for 24 h, lower than the mass loss of the fibers treated with all three NaOH concentrations for 48 h, and only slightly lower than the mass loss of the fibers treated with 10% NaOH for 72 h. This can be associated with a higher percentage of the hemicellulose content that allows the fibers to retain more moisture [41]. The NaOH mercerization eliminated some proportion of hemicellulose and lignin; thus, the fibers tended to absorb less moisture on their surface [42].

The T_onset_ values found for 24, 48 and 72 h of the sedge-treated fibers were slightly lower when compared to those of the untreated fibers. As for the maximum rate, all treatments presented to be slightly higher, indicating higher thermal stability at higher temperatures. Furthermore, it could be observed that in all treated fiber concentrations the hemicellulose content was decomposed within a range of 240–360 °C. This behavior is in accordance with other reports in the literature [41].

The sudden mass drop starting at the T_onset_ region resulted in mass losses around ~58% for all NaOH concentrations of 3%, 5%, and 10%. The untreated sedge fiber showed a slightly higher mass loss, indicating that the treated fibers presented better thermal stability at higher temperatures. The third stage ranged from 300 to 590 °C for all the treated sedge fibers and resulted in a total mass loss around ~95% for the 3%, 5%, and 10% NaOH concentrations. The final mass loss occurred at higher temperatures for the treated fibers when compared to that found for the untreated sedge fibers at 496 °C. Further, after this range, no degradation and mass loss occurred until the end of the test at 600 °C, due to the ash content present.

It is extensively known that the final stage of degradation is directly associated with the elimination of lignin, within a range of 560–580 °C for all conditions. Observing all the treatments for different durations of time, it can be noted that the treatments related to 3% NaOH for 24 h and 48 h exhibited better thermal stability properties. Furthermore, the group treated with 3% NaOH for 48 h showed that the lignin present did not fully decompose. For this reason, it can be selected for its use as a reinforcement for composites. Furthermore, previous results incorporating untreated sedge as a reinforcement for an epoxy matrix showed that the composites disclose poor thermal stability when compared to the neat epoxy [34]. This reason is due to the fact that natural fibers present a lower thermal stability and influence the thermal behavior of the composites. Therefore, it is expected that the novel composites reinforced by alkali-treated sedge present a higher thermal stability. 

### 3.2. DSC Analysis

DSC plots of treated sedge fibers shown in Figure 4 helped to support the degradation of cellulose compound previously witnessed in TGA curves. The endothermic peak noticed ranging from 69 to 75 °C is related to the thermal energy required for the vaporization content present in the fiber surface [42]. Then, it was observed that the exothermic peak ranging from 334 to 334 °C relates to the degradation of α cellulose and hemicellulose [43]. Further, the curves continued to travel in the endothermal region beyond 400 °C due to the volatilization of lignin and wax. As usual, the lignin compound in fiber started to decompose at a temperature range of 280 to 520 °C [44], which was observed in this thermal event on the endothermal regions of the curves. At temperatures beyond 600 °C, the curves continued to swing toward the exothermic region due to the breakdown of some chemical compounds in hemicellulose and lignin deposits [42].

The endothermic peak noticed ranging from 81 to 92 °C in Figure 5 is related to the thermal energy required for the vaporization content present in the fiber surface [42]. Further, the curves continued to travel in the endothermal region up to the temperature range of 256–286 °C due to the volatilization of lignin and wax. Then, it was observed that the exothermic peak ranging from 338 to 354 °C related to the degradation of α cellulose and hemicellulose [43]. At temperatures beyond 400 °C, only the curve under a 10% NaOH solution condition continued to swing toward the exothermic region due to the breakdown of some chemical compounds in hemicellulose and lignin [42].

The endothermic peak noticed ranging from 69 to 72 °C in Figure 6 is related to the thermal energy required for the vaporization content present in the fiber surface [42]. Further, the curves continued to travel in the endothermal region up to the temperature range of 261–274 °C due to the volatilization of lignin and wax. Then, it was observed that the exothermic peak ranging from 331 to 362 °C related to the degradation of α cellulose and hemicellulose [43]. At temperatures beyond 400 °C, the curves continued to swing toward the exothermic region due to the breakdown of some chemical compounds in hemicellulose and lignin deposits [42].

Comparing all the tests, the sedge fibers treated with NaOH at a concentration of 3% obtained low values of the thermal curves when compared with those treated with 5% and 10% NaOH. In addition, the untreated fibers showed lower values than those with the 3% NaOH treatment [38]. Under the time of 48 h, better thermal stability with a decay of the thermal curves was shown for all treatment conditions.

### 3.3. FTIR Analysis

Figure 7, Figure 8 and Figure 9 illustrate the FTIR spectra of NaOH-treated sedge fibers within the range of 4000–400 cm^−1^. The region at 3453−3489 cm^−1^ indicates bands of O-H functional groups in cellulose. The band at 2915−2915 cm^−1^ is identified as the C-H stretching of cellulose [45], and the peak at 2366 cm^−1^ is assigned to the bond between organic molecules and wax compounds in the fiber surface [42]. The common peak under all conditions at 1641 cm^−1^ is related to C=O aromatic groups in the lignin component present in the sedge fibers [46]. The band at 1381−1381 cm^−1^ can be related to the CH3 symmetric bending according to [47]. The band at 1119 cm^−1^ is associated with C-O stretching vibrations, and the band at 674 cm^−1^ is related to C-C stretching. It could be stated that the treatment in the conditions investigated was not enough to reduce the content of hemicellulose and lignin, although the peak related to wax content was absent at the 10% NaOH treatment.

In Figure 8, the region at 3452−3482 cm^−1^ indicated bands of O-H functional groups in cellulose. The band at 2361−2367 cm^−1^ is assigned to the bond between organic molecules and wax compounds in the fiber surface, which is almost absent for the 5% NaOH condition [42]. The peak at 1638−1649 cm^−1^ is related to C=O aromatic groups in lignin and hemicellulose components present in the sedge fibers [46]. It can be noted that the treatment for 24 h resulted in a strong presence of lignin and hemicellulose, which was shown in lower NaOH concentrations for 48 h. The band at 1384−1390 cm^−1^ can be related to the CH3 symmetric bending according to [47]. The band at 1119−1125 cm^−1^ is associated with C-O stretching vibrations, and the band at 664−955 cm^−1^ is related to the C-C stretching. It could be stated that the treatment in the conditions investigated was not enough to reduce the content of hemicellulose and lignin, although the conditions of 24 h presented stronger peaks related to the C=O aromatic groups.

In Figure 9, the region at 3456−3483 cm^−1^ indicated bands of O-H functional groups in cellulose. The band at 2921−2927 cm^−1^ is identified as the C-H stretching of cellulose [45], and the peak at 2360 cm^−1^ is assigned to the bond between organic molecules and wax compounds in the fiber surface [42]. The peak at 1644 cm^−1^ is related to C=O aromatic groups in lignin and hemicellulose components present in the sedge fibers [46]. It can be noted that the conditions treated for 24 h had a strong presence of lignin and hemicellulose, which was shown in lower concentrations for 72 h. The band at 1381−1386 cm^−1^ can be related to the CH3 symmetric bending according to [47]. The bands at 1111−1117 cm^−1^ and 1034 cm^−1^ are associated with C-O stretching vibrations, and the band at 664−955 cm^−1^ is related to the C-C stretching. It could be stated that the treatment in the conditions investigated was not enough to reduce the content of hemicellulose and lignin, although the conditions of 24 h presented stronger peaks related to the C=O aromatic groups.

For all treatments of different percentages and times, the disappearance of the 1731 cm^−1^ band stood out in comparison with the untreated fiber [35]. This band is associated with the stretching of ester and carboxyl groups (C-O compound) present in hemicellulose; this band vanished during the NaOH treatment, indicating the removal of hemicellulose [42].

### 3.4. XRD Analysis

Table 3 highlights the peaks, the crystallite sizes, and the crystallinity indices of the untreated and treated sedge fibers. The peak (1 1 0) is attributed to a reflection of the crystalline structure in cellulose I, while the halo of (1 0 1) reflection is associated with the amorphous region composed of lignin and hemicellulose structures [42]. The reflection of (0 0 2) is related to the cellulose IV fraction [45]. After the alkali treatment, it can be noticed that for some conditions the peak at (1 1 0) appeared. However, the amorphous halo was present. This might indicate that a significant amount of lignin and hemicellulose remained, although they were partially removed. In addition, this condition was the only one presenting an increase in the fraction of crystalline cellulose, as shown in the diffractogram in Figure 10, due to the removal of low-molecular-weight hemicellulose, wax, and lignin compounds as a result of NaOH treatment [45]. On the other hand, under all conditions, an increase in the crystallite size was presented, which was caused by the swelling of the cellulose chain with Na. It also penetrated the crystal lattice and reacted with the OH group after breaking the hydrogen bonds and producing the cellulose−alkali complex [48]. Although the increase in crystallite size favored resistance to moisture absorption and chemical reactivity of sedge fibers [42], a higher concentration of NaOH could lower the content of α cellulose since strong degradation occurred by intracrystalline swelling [48]. This could be the reason why most conditions had a lower crystallinity index. The results in Table 3 were in contrast with this hypothesis, because all the other NLFs presented an increase in the crystallinity index as a function of crystallite size. Possibly in the case of the sedge fibers, the treatment might be too strong, so it damaged the microstructure.

### 3.5. Morphological Analysis

SEM images for the alkali-treated fibers are shown in Figure 11 and Figure 12. Figure 11 depicts the untreated and treated fibers with 3% NaOH in different periods of time. It is clear that not much hemicellulose, wax, impurities, and lignin existed in Figure 11a,b. These contents impaired the interfacial bonding characteristics of the fiber during composite formation. In contrast, in micrographs in Figure 11d–f, it is possible to visualize the amorphous content on the surface of the fibers. Additionally, lignin and hemicellulose were bonded as glues on a microfibril in Figure 11e. In other words, this amorphous content played a role of a matrix phase for this natural biocomposite [14]. Another evidence that the alkali treatment modified the fibers is the void present in Figure 11b. It was occasioned by the removal of the amorphous phase, indicating the strength of the fibers might reduce significantly. 

Figure 12 is related to all 5% and 10% alkali-treated sedge fibers surface. Fibers treated with 72, 48, and 24 h of the immersion time presented some degradation aspects on their surface, proving that the treatment was so strong that it damaged the fiber structure. This might reduce their mechanical properties and negatively impact the performance of the composites. Surprisingly, the conditions related to a 10% NaOH did not follow the same degradation behavior. Despite that, the fibers showed hollow spaces along their longitudinal length. Once again, it may be assigned to a greater removal extent in comparison to the condition presented in Figure 11. Therefore, under these conditions of 5% and 10% NaOH, damages on the surface of the fiber and greater extension of voids were present. This could reduce the mechanical performance when acting as a reinforcement phase. For this reason, the lowest (3%) NaOH concentration could lead to a possible enhancement for composites reinforced by this fiber.

## 4. Conclusions

The work evaluated the effect of chemical treatment with NaOH on natural sedge fibers. Different concentrations of immersion with 3%, 5%, and 10% of NaOH with different times of 24, 48, and 72 h presented parameters that can be applied for obtaining better properties of the sedge fiber in polymeric composites.

TGA/DTG analysis revealed mass loss type results up to 200 °C, at the end of the second stage, and up to 600 °C, in addition to the initial degradation and the maximum degradation rate. Comparing all the tests, the sedge fibers treated with NaOH at a concentration of 3% obtained the lower values of the thermal curves when compared with those under 5% and 10% NaOH treatments. In addition, the untreated fibers showed lower values than those with the 3% NaOH treatment. Under the time of 48 h, better thermal stability with a decay of the thermal curves was shown for all treatment conditions.

DSC analysis showed the endothermic and exothermic peaks that better clarified the degradation of cellulose and hemicellulose and volatilization of lignin and wax present in the structure of the sedge fiber. For all treatments of different concentrations and times, the disappearance of the 1731 cm^−1^ band stood out in comparison with the untreated fiber. This band is associated to the stretching of ester and carboxyl groups (C-O compound) present in hemicellulose. The increase in crystallite size favored resistance to moisture absorption and chemical reactivity of sedge fibers; a higher concentration of NaOH could lower the content of α cellulose, since strong degradation occurred by intracrystalline swelling.

SEM images for the fibers treated with alkali showed that there was not as much presence of hemicellulose, wax, impurities, and lignin as shown for the fibers treated with 3% NaOH in different periods of time. Under conditions of 5% and 10% NaOH, damages on the surface of the fiber and greater extension of voids were present. For this reason, the lowest (3%) NaOH concentration has the potential to enhance composites reinforced by this fiber. 

## Figures and Tables

**Figure 1 polymers-15-02153-f001:**
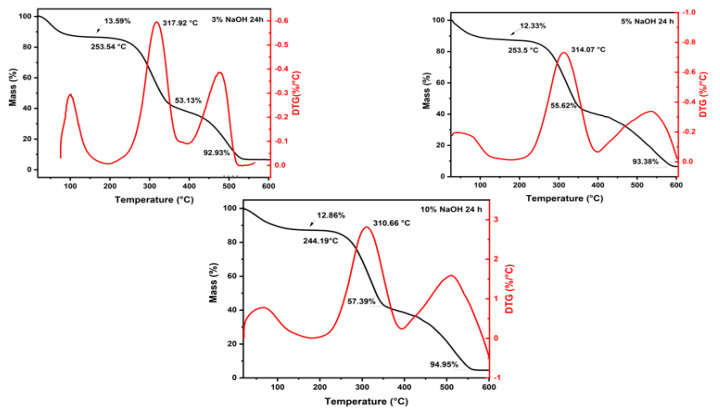
TGA and DTG curves for all the concentrations immersed in a NaOH solution for 24 h.

**Figure 2 polymers-15-02153-f002:**
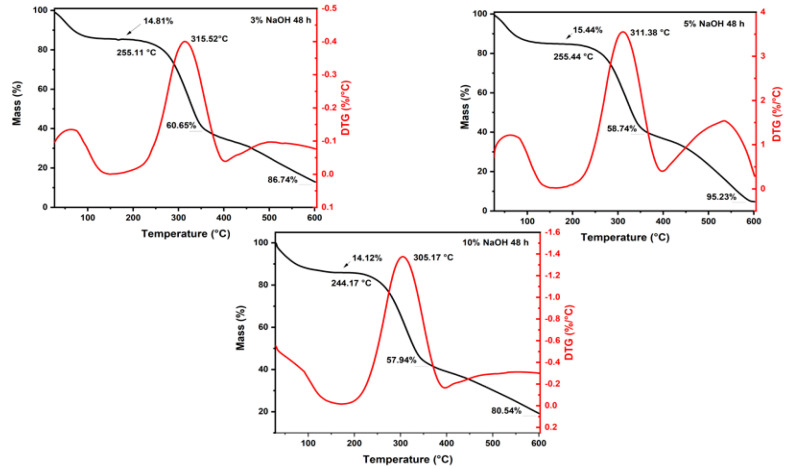
TGA and DTG curves for all the concentrations immersed in a NaOH solution for 48 h.

**Figure 3 polymers-15-02153-f003:**
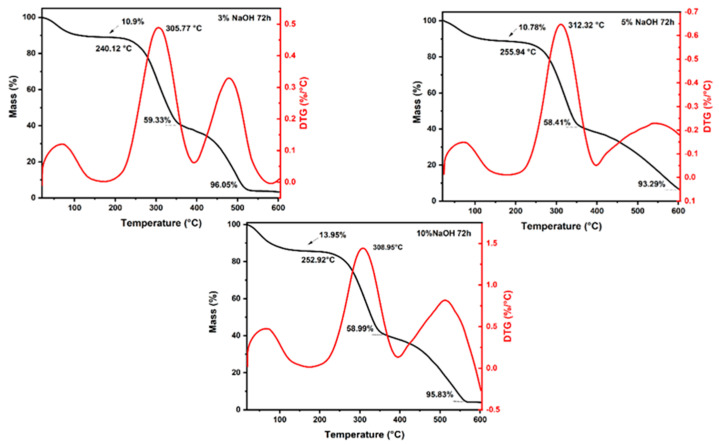
TGA and DTG curves for all the concentrations immersed in a NaOH solution for 72 h.

**Figure 4 polymers-15-02153-f004:**
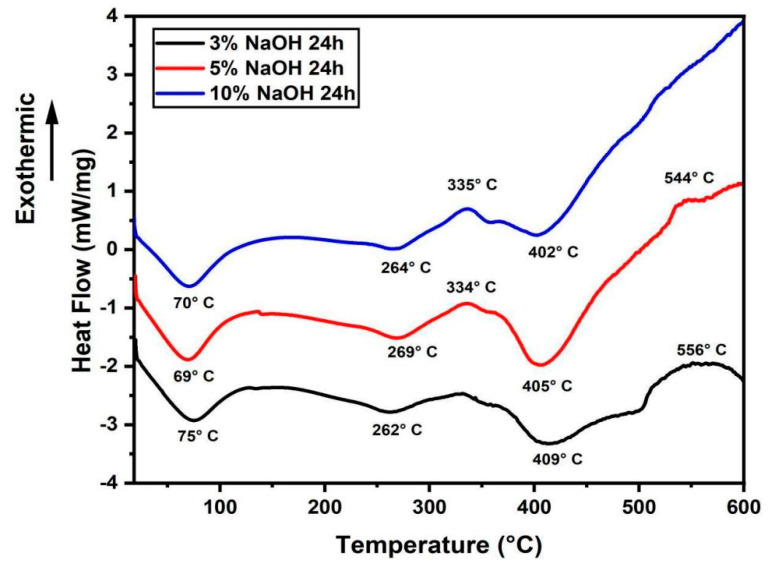
DSC curves for all the concentrations immersed in a NaOH solution for 24 h.

**Figure 5 polymers-15-02153-f005:**
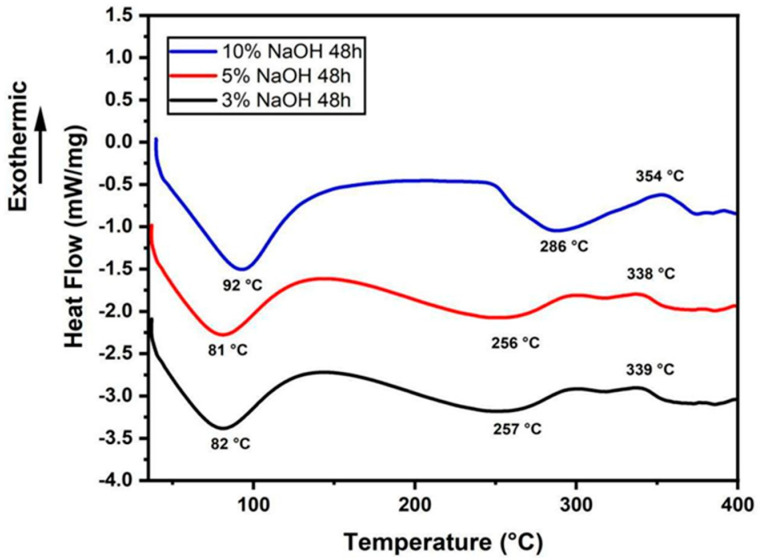
DSC curves for all the concentrations immersed in a NaOH solution for 48 h.

**Figure 6 polymers-15-02153-f006:**
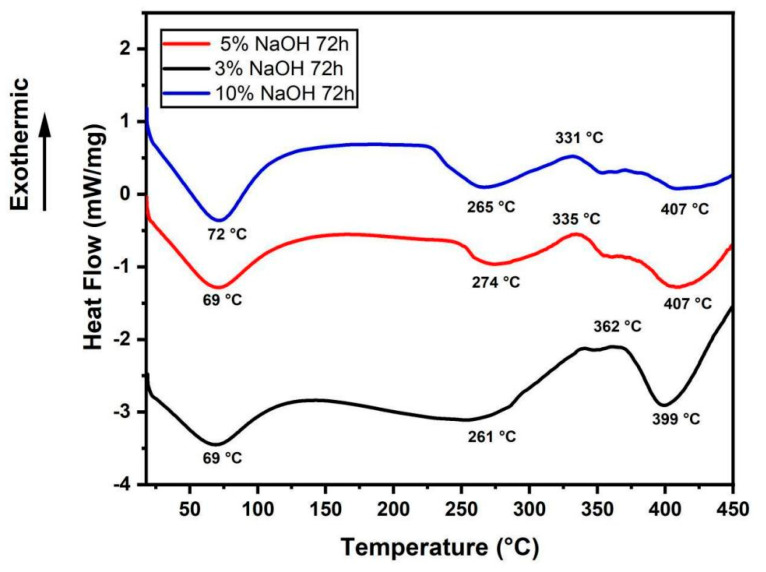
DSC curves for all the concentrations immersed in a NaOH solution for 72 h.

**Figure 7 polymers-15-02153-f007:**
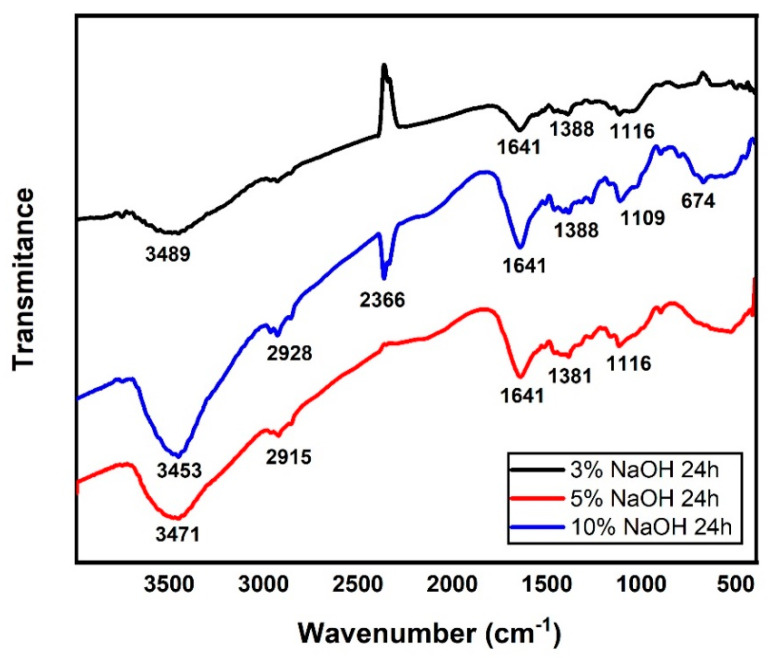
FTIR spectra for all the concentrations immersed in a NaOH solution for 24 h.

**Figure 8 polymers-15-02153-f008:**
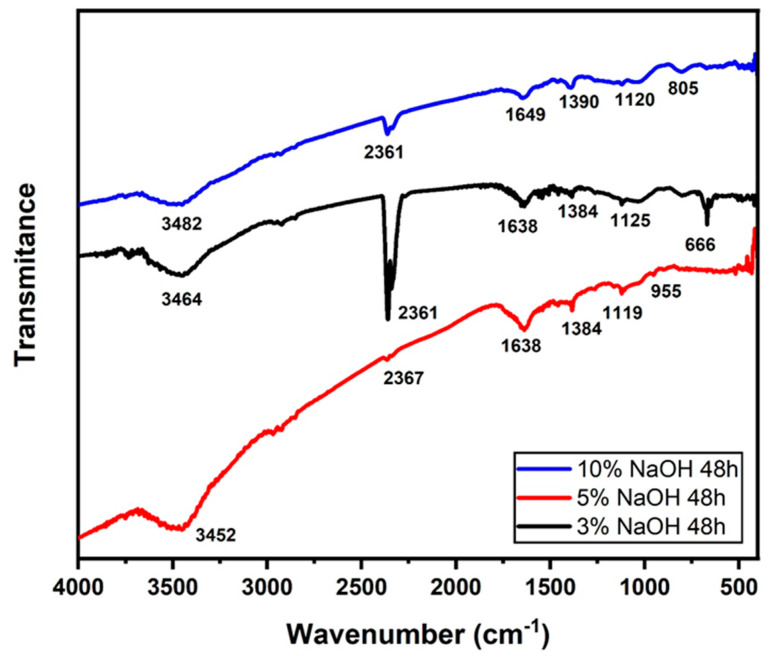
FTIR spectra for all the concentrations immersed in a NaOH solution for 48 h.

**Figure 9 polymers-15-02153-f009:**
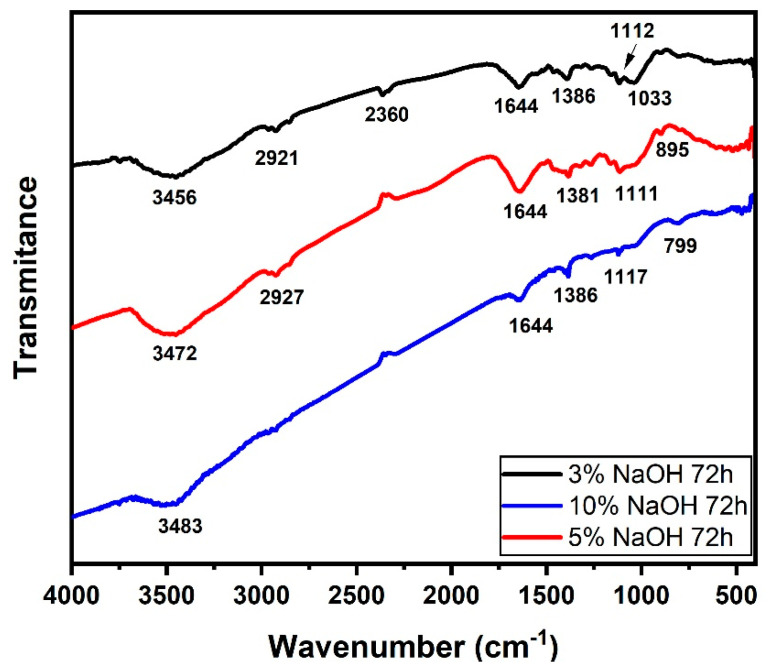
FTIR spectrum for all the concentrations immersed in a NaOH solution for 72 h.

**Figure 10 polymers-15-02153-f010:**
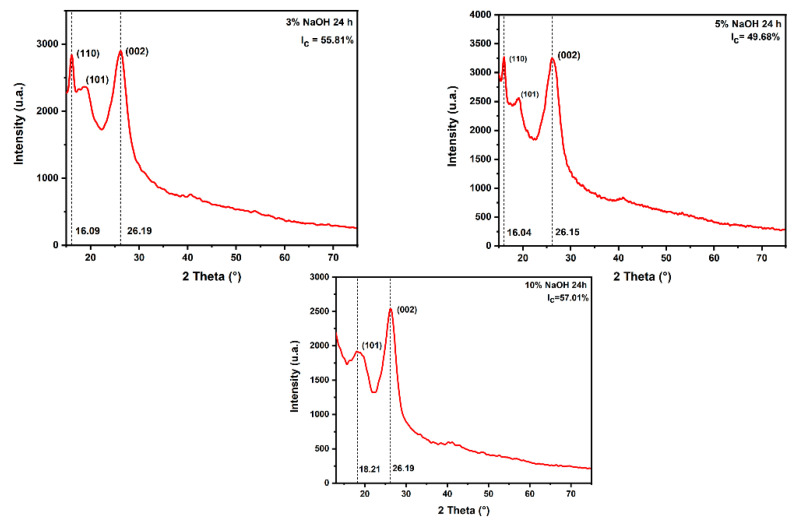
XRD diffractograms regarding a concentration immersed in a 3% NaOH solution for all periods of time.

**Figure 11 polymers-15-02153-f011:**
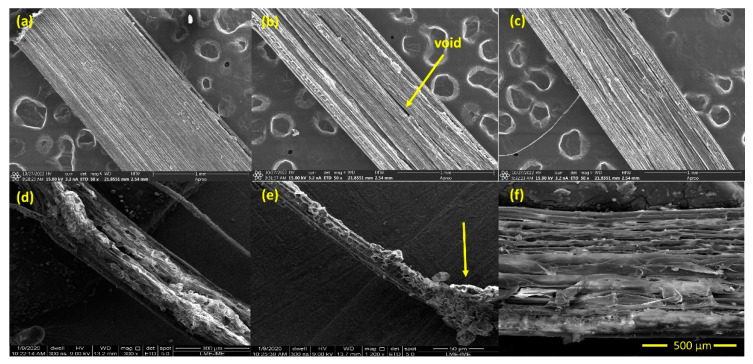
SEM images of alkali-treated fibers in a NaOH concentration of 3% for 24 h (**a**), 48 h (**b**), and 72 h (**c**) immersion periods; SEM images of untreated sedge fibers for 24 h (**d**), 48 h (**e**), and 72 h (**f**) immersion periods.

**Figure 12 polymers-15-02153-f012:**
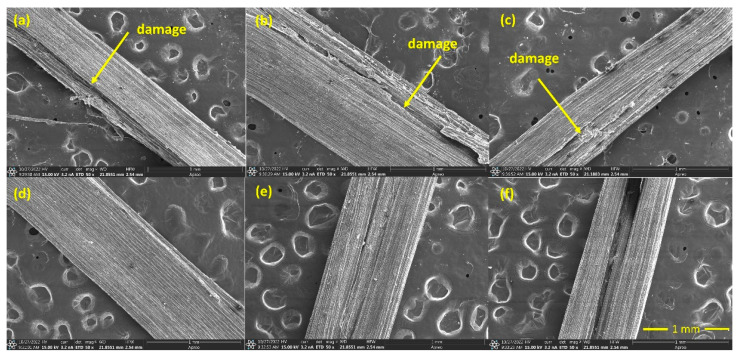
SEM images of alkali-treated fibers in a concentration of 5% NaOH for the immersion periods of 24 h (**a**), 48 h (**b**), and 72 h (**c**); SEM images of alkali-treated fibers in a concentration of 10% NaOH for the immersion times of 24 h (**d**), 48 h (**e**), and 72 h (**f**).

**Table 1 polymers-15-02153-t001:** Alkali conditions based on the time and NaOH concentration.

Parameters			
NaOH Concentration	Immersion Period		
-	24 h	48 h	72 h
3%	Group 1	Group 4	Group 7
5%	Group 2	Group 5	Group 8
10%	Group 3	Group 6	Group 9

**Table 2 polymers-15-02153-t002:** Thermogravimetric parameters for all the concentrations immersed in a NaOH solution for 24, 48, and 72 h.

Conditions		Mass Loss			Initial Degradation (°C)	Maximum Degradation Rate (°C)
		Up to 200 °C (%)	At the End of the Second Stage (%)	At 600 °C (%)		
24 h	3%	13.59	53.13	92.93	253.54	317.92
	5%	12.33	55.62	93.38	253.5	314.07
	10%	12.86	57.39	94.95	244.19	310.66
48 h	3%	14.81	60.65	86.74	255.11	315.52
	5%	15.44	58.74	95.23	255.44	311.38
	10%	14.12	57.94	80.54	244.17	305.17
72 h	3%	10.9	59.33	96.05	240.12	305.77
	5%	10.78	58.41	93.29	255.94	312.32
	10%	13.95	58.99	95.83	252.92	308.95
	Untreated *	13.23	63.07	96.04	255.68	300.95

* Neuba et al. [38].

**Table 3 polymers-15-02153-t003:** Crystallite sizes and crystallinity indices of the untreated and alkali-treated sedge fibers, as well as other NLFs.

Treatments		Crystallinity Index (%)	Crystallite Size (nm)	Reference
Raw *Pongamia pinnata* L.		45.31	5.43	[49]
Alkalized *Pongamia pinnata* L.		52.43	8.43	[49]
Raw banyan Fiber		72.47	6.28	[50]
Alkalized banyan Fiber		76.35	7.74	[50]
Raw ACF		27.5	4.17	[45]
Alkalized ACF		35.6	6.43	[45]
Untreated sedge fibers		62.47	2.56	[51]
**Treatments with Different NaOH Concentrations**		**Crystallinity** **Index (%)**	**Crystallite Size (nm)**	**Reference**
24 h	3%	55.81	3.75	Present work
	5%	49.68	3.43	
	10%	57.01	3.44	
48 h	3%	66.50	10.66	
	5%	51.24	4.09	
	10%	39.40	6.84	
72 h	3%	57.21	3.66	
	5%	52.71	3.68	
	10%	56.4	3.89	

## Data Availability

Not applicable.

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
