# Peer review of "Alkaline Treatment Investigation for Sedge Fibers (Cyperus malaccensis): A Promising Enhancement"

_polymers, 2023, doi:10.3390/polym15092153_

Round 1

Reviewer 1 Report

Review: polymers-2364712

Title: Alkaline Treatment Investigation for Sedge Fibers (Cyperus malaccensis): A Promising Enhancement

The study presents a comparative analysis of the physical and chemical changes produced by the nine types of treatments applied to sedge fibers (Cyperus malaccensis).

The presented studies are valuable for establishing a chemical treatment technology for sedge fibers in order to improve their physical and chemical properties.

The article is interesting, and I recommend publication.

Small changes should be made by the authors:

·       1)  Pag. 3, line 123 – explain the GO notation, when it appears for the first time in the text;

·    2) Figures 1 - 3 can be arranged in a matrix so that the concentration is on the columns and the exposure time is on the rows.

·      3) why is the chemical fingerprint slightly different for the samples immersed in NaOH for 48 hours (Fig. 8), compared to those immersed for 24 and 72 hours (Figs. 7 and 9)?

Reviewer 2 Report

The manuscript demonstrated the alkaline treatment investigation for Sedge Fibers (Cyperus malaccensis). The effect of NaOH concentration and immersion time on the thermal behavior, chemical structure, physical structure and morphological aspect of the treated fiber was highlighted. In the analysis of the experimental results, related literature has been cited to increase the credibility of experimental results. Overall, the conclusion was basically supported by the findings. However, there are still some issues that need to be addressed prior to publication.

1. In the “Introduction section”, the novelty of the proposed concept should be emphasized and improved. For instance, the reason why the seven-islands-sedge were selected for the preparation of natural lignocellulosic fibers should be well commented.

2.  There are two “Group 4” in Table 1.

3.  Like the infrared analysis data, XRD analysis should include all the concentrations immersed in a NaOH solution for 24 h.

4. In Figure 7,8 and 9, the transmittance of FITR is relative value, the unit % is not accurate.

5. The scale-bar should be added to Fig.11-12

6. The conclusion should be shortened and improved to highlight the main results.

7. “A Promising Enhancement” concept should be demonstrated in the experimental findings. Related results regarding the mechanical reinforcement should be added to the revised manuscript.

The manuscript demonstrated the alkaline treatment investigation for Sedge Fibers (Cyperus malaccensis). The effect of NaOH concentration and immersion time on the thermal behavior, chemical structure, physical structure and morphological aspect of the treated fiber was highlighted. In the analysis of the experimental results, related literature has been cited to increase the credibility of experimental results. Overall, the conclusion was basically supported by the findings. However, there are still some issues that need to be addressed prior to publication.

1. In the “Introduction section”, the novelty of the proposed concept should be emphasized and improved. For instance, the reason why the seven-islands-sedge were selected for the preparation of natural lignocellulosic fibers should be well commented.

2.  There are two “Group 4” in Table 1.

3.  Like the infrared analysis data, XRD analysis should include all the concentrations immersed in a NaOH solution for 24 h.

4. In Figure 7,8 and 9, the transmittance of FITR is relative value, the unit % is not accurate.

5. The scale-bar should be added to Fig.11-12

6. The conclusion should be shortened and improved to highlight the main results.

7. “A Promising Enhancement” concept should be demonstrated in the experimental findings. Related results regarding the mechanical reinforcement should be added to the revised manuscript.

Round 2

Reviewer 2 Report

In general, the revision work is extensive and convincing. Therefore, this reviewer would like to recommend the pape for publication.